# BOOMERANG: LOCAL SAMPLING ON IMAGE MANIFOLDS USING DIFFUSION MODELS

## ABSTRACT

Diffusion models can be viewed as mapping points in a high-dimensional latent space onto a low-dimensional learned manifold, typically an image manifold. The intermediate values between the latent space and image manifold can be interpreted as noisy images which are determined by the noise scheduling scheme employed during pre-training. We exploit this interpretation to introduce Boomerang, a local image manifold sampling approach using the dynamics of diffusion models. We call it Boomerang because we first add noise to an input image, moving it closer to the latent space, then bring it back to the image space through diffusion dynamics. We use this method to generate images which are similar, but nonidentical, to the original input images on the image manifold. We are able to set how close the generated image is to the original based on how much noise we add. Additionally, the generated images have a degree of stochasticity, allowing us to locally sample as many times as we want without repetition. We show three applications for which Boomerang can be used. First, we provide a framework for constructing privacy-preserving datasets having controllable degrees of anonymity. Second, we show how to use Boomerang for data augmentation while staying on the image manifold. Third, we introduce a framework for image super-resolution with 8x upsampling. Boomerang does not require any modification to the training of diffusion models and can be used with pretrained models on a single, inexpensive GPU.

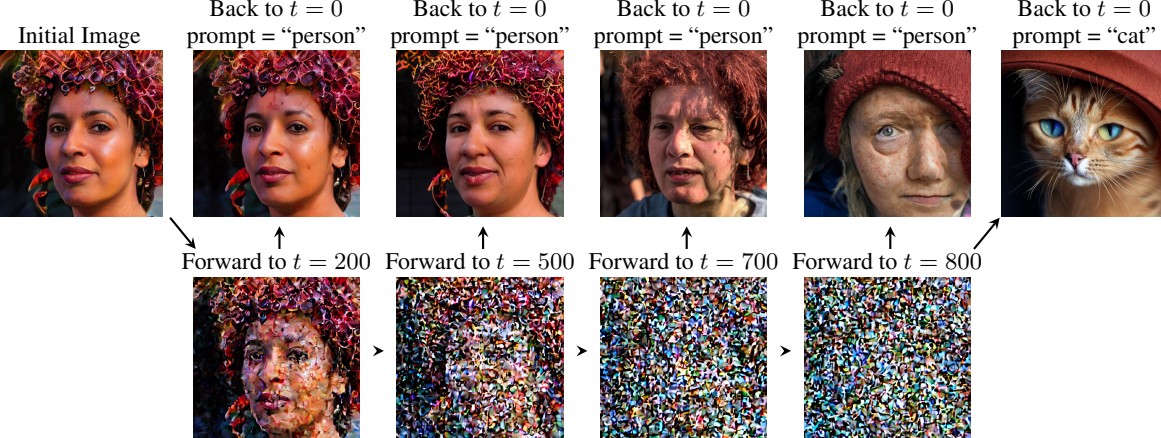

Figure 1: Boomerang via Stable Diffusion (Rombach et al., 2022). Code available here. Starting from an initial image $\mathbf{x}_0 \sim p(\mathbf{x}_0)$, we add varying levels of noise to the latent variables according to the noise schedule of the forward diffusion process. Boomerang maps the noisy latent variables back to the image manifold by running the reverse diffusion process starting from the reverse step associated with the added noise. The resulting images are local samples from the image manifold, where the closeness is determined by the amount of added noise. While Boomerang here is applied to the Stable Diffusion model, it is applicable to other types of diffusion models, e.g., denoising diffusion models (Ho et al., 2020). Additional images are provided in Appendix A.1.

# 1 INTRODUCTION

Generative models have seen a tremendous rise in popularity and applications over the past decade, ranging from image synthesis, audio generation, out-of-distribution data detection, and reinforcement learning to drug synthesis (Kingma & Welling, 2014; Goodfellow et al., 2014; Bond-Taylor et al., 2021). One of the key benefits of generative models is that they can generate new samples from an unknown probability distribution from which we have samples. Recently, with the advent of models such as Dall-E 2 (Ramesh et al., 2022), Imagen (Saharia et al., 2022a), and Stable Diffusion (Rombach et al., 2022), a family of generative models known as diffusion models have gained attention in both the academic and public spotlights. A key difference between diffusion models and previous state-of-the art generative models, such as Generative Adversarial Networks (GANs) (Goodfellow et al., 2014), is that diffusion models take a fundamentally different approach to image synthesis, opting for a series of denoising objectives, whereas GANs opt for saddle-point objectives that have proven to be difficult to train (Yadav et al., 2018; Mescheder et al., 2018; Dhariwal & Nichol, 2021).

Generative models estimate the underlying probability distribution, or manifold, of a dataset. As such, generative models should be able to produce samples close to a particular data point—known as local sampling on the manifold—-given that they have some knowledge about properties of the data manifold. One application of local sampling is to remove noise from an image, especially in the cases of severe noise, where traditional denoising methods might fail (Kawar et al., 2021). Another application of local sampling is data augmentation, which involves applying transformations onto copies of data points to produce new data points. These data points are still from the original data distribution, but different enough from existing data to encourage the model to generalize in downstream tasks such as classification (Wong et al., 2016). While established techniques in image augmentation include crops, flips, rotations, and color manipulations, data augmentation techniques for images and other data types are an ongoing field of research (Yang et al., 2022; Wen et al., 2021; Feng et al., 2021).

Diffusion models are typically designed to sample globally on the image manifold instead of performing local sampling. GANs (Goodfellow et al., 2014), VAEs (Kingma & Welling, 2014), and NFs (Rezende & Mohamed, 2015) can sample locally to a limited extent. VAEs and NFs can project a data point $\mathbf{x}$ into a latent vector $\mathbf{z}$ in their respective latent spaces and then re-project $\mathbf{z}$ back into the original data space, producing an estimate $\mathbf{x}'$ of the original data point. The key difference between $\mathbf{x}'_{\mathrm{VAE}}$ and $\mathbf{x}'_{\mathrm{NF}}$ is that neither the encoder nor the decoder in a VAE is invertible, while the projection function of an NF is invertible, resulting in $\mathbf{x}'_{\mathrm{VAE}} \approx \mathbf{x} = \mathbf{x}'_{\mathrm{NF}}$ (Kobyzev et al., 2021). Meanwhile, GANs generally do not learn a map from the data space to the latent space, instead opting to only learn a map from the latent space to the data space; finding or training the best methods for GAN inversion is an ongoing area of research (Karras et al., 2020; Xia et al., 2022). Previous work on GAN inversion has shown that, without special attention to underrepresented data points during GAN training, reconstruction of certain data points can fail (Yu et al., 2020). As such, VAEs and NFs can perform local sampling of a data point $\mathbf{x}$ by projecting a perturbed version of its corresponding latent $\mathbf{z}$ back into the original data space, while GANs also have the potential to do so, albeit requiring a suitable GAN inversion method (Wang et al., 2022; Zhu et al., 2020). While the straightforward tractability of VAE- or NF-based local sampling is attractive, GANs currently outperform VAEs and NFs on popular tasks such as image synthesis (Zhang et al., 2022; Sauer et al., 2022). However, given the recent advent and popularity of diffusion models and the brittle nature of GAN training (Saxena & Cao, 2021), local sampling using diffusion models is a promising avenue for leveraging local sampling techniques towards problems such as anonymization, data augmentation, and super-resolution.

We propose the Boomerang algorithm to enable local sampling of image manifolds using diffusion models. The algorithm earns its name from its principle mechanism—using noise of a certain variance to push data away from the image manifold, and then using a diffusion model to pull the noised data back onto the manifold. The variance of the noise is the only parameter in the algorithm, and governs how similar the new image is to the old image, as reported by reported by Ho et al. (2020). We show how this technique can be used within three applications: (1) data anonymization for privacy-preserving machine learning; (2) data augmentation; and (3) image super-resolution. We show that by exploiting the proposed local sampling technique we are able to anonymize the dataset and maintain better classification accuracy when compared with state-of-the-art generated data from

StyleGAN-XL (Sauer et al., 2022). For data augmentation, we obtain higher classification accuracy when trained on the Boomerang-augmented dataset versus no augmentation at all. Finally, we show that Boomerang can be used for super-resolution at varying strengths depending on the desired upsampling factor without the need to train different networks.

In Section 2 we discuss the training framework of diffusion models, introducing the forward and reverse processes. In Section 3 we introduce our proposed local sampling method—Boomerang—and provide insights on how the amount of added noise affects the locality of the resulting samples. Finally, we describe three applications (Sections 4 to 6) that Boomerang can be used without any modification to the diffusion pretraining.

## 2 DIFFUSION MODELS

Generative models aim to sample from a target probability distribution by using available samples from the distribution as training data, e.g., images of human faces. Diffusion models—a class of generative models—accomplish this by learning to reverse a diffusion (forward) process, which involves adding Gaussian noise to the input image in $T$ steps (Ho et al., 2020). Given an image from the target distribution $\mathbf{x}_0 \sim p(\mathbf{x}_0)$, the *forward process* can be written as:

$$\mathbf{x}_t := \sqrt{1 - \beta_t}\mathbf{x}_{t-1} + \boldsymbol{\epsilon}_t, \quad \boldsymbol{\epsilon}_t \sim \mathcal{N}(\mathbf{0}, \beta_t\mathbf{I}), \quad t = 1, \dots, T, \tag{1}$$

where $\beta_t \in (0, 1)$, $t = 1, \dots, T$, is the noise variance at step $t$, which is typically chosen beforehand (Song & Ermon, 2020). Since the transition from step $t-1$ to $t$ is defined by a Gaussian distribution in the form of $q(\mathbf{x}_t|\mathbf{x}_{t-1}) := \mathcal{N}(\sqrt{1 - \beta_t}\mathbf{x}_{t-1}, \beta_t\mathbf{I})$, the distribution of $\mathbf{x}_t$ conditioned on the clean input image $\mathbf{x}_0$ can be expressed as a Gaussian distribution,

$$q(\mathbf{x}_t|\mathbf{x}_0) = \mathcal{N}\left(\sqrt{\alpha_t}\mathbf{x}_0, (1 - \alpha_t)\mathbf{I}\right), \quad t = 1, \dots, T, \tag{2}$$

with $\alpha_t = \prod_{i=1}^{t}(1 - \beta_i)$. During training, diffusion models learn to reverse the forward process by starting at $t = T$ with a sample from the standard Gaussian distribution $\mathbf{x}_T \sim \mathcal{N}(\mathbf{0}, \mathbf{I})$. The *reverse process* is defined via a Markov chain over $\mathbf{x}_0, \mathbf{x}_1, \dots, \mathbf{x}_T$ such that

$$\mathbf{x}_{t-1} := f_\phi(\mathbf{x}_t, t) + \boldsymbol{\eta}_t, \quad \boldsymbol{\eta}_t \sim \mathcal{N}(\mathbf{0}, \bar{\beta}_t\mathbf{I}), \quad t = 1, \dots, T. \tag{3}$$

In the above expression, $f_\phi(\mathbf{x}_t, t)$ is parameterized by a neural network with weights $\phi$, and $\bar{\beta}_t\mathbf{I}$ denotes the covariances at step $t$. Equation (3) represents a chain with transition probabilities defined with Gaussian distributions with density

$$p_\phi(\mathbf{x}_{t-1} \mid \mathbf{x}_t) := \mathcal{N}(f_\phi(\mathbf{x}_t, t), \bar{\beta}_t\mathbf{I}), \quad t = 1, \dots, T. \tag{4}$$

The covariance $\bar{\beta}_t\mathbf{I}$ in different steps can be also parameterized by neural networks, however, we follow Luhman & Luhman (2022) and choose $\bar{\beta}_t = \frac{1-\alpha_{t-1}}{1-\alpha_t}\beta_t$, that matches the forward posterior distribution when conditioned on the input image $q(\mathbf{x}_{t-1} \mid \mathbf{x}_t, \mathbf{x}_0)$ (Ho et al., 2020).

To ensure the Markov chain in Equation (3) reverses the forward process (Equation 1), the parameters $\phi$ are updated such that the resulting image at step $t = 0$ via the reverse process represents a sample from the target distribution $p(\mathbf{x}_0)$. This can be enforced by maximizing—with respect to $\phi$—the likelihood $p_\phi(\mathbf{x}_0)$ of training samples where $\mathbf{x}_0$ represents the outcome of the reverse process at step $t = 0$. Unfortunately, the density $p_\phi(\mathbf{x}_0)$ does not permit a closed-form expression. Instead, given the Gaussian transition probabilities defined in Equation (4), the joint distribution over all the $T + 1$ states can be factorized as,

$$p_\phi(\mathbf{x}_0, \dots, \mathbf{x}_T) = p(\mathbf{x}_T) \prod_{t=1}^{T} p_\phi(\mathbf{x}_{t-1} \mid \mathbf{x}_t), \quad p_T(\mathbf{x}_T) = \mathcal{N}(\mathbf{0}, \mathbf{I}), \tag{5}$$

with all the terms on the right-hand-side of the equality having closed-form expressions. To obtain a tractable expression for training diffusion models, we treat $\mathbf{x}_1, \dots, \mathbf{x}_T$ as latent variables and use the negative evidence lower bound (ELBO) expression for $p_\phi(\mathbf{x}_0)$ as the loss function,

$$\mathcal{L}(\boldsymbol{\phi}) := \mathbb{E}_{p(\mathbf{x}_0)}\mathbb{E}_{q(\mathbf{x}_1, \dots, \mathbf{x}_T|\mathbf{x}_0)}\left[-\log p_T(\mathbf{x}_T) - \sum_{t=1}^{T} \log \frac{p_\phi(\mathbf{x}_{t-1} \mid \mathbf{x}_t)}{q(\mathbf{x}_t|\mathbf{x}_{t-1})}\right] \tag{6}$$

$$\geq \mathbb{E}_{p(\mathbf{x}_0)}\left[-\log p_\phi(\mathbf{x}_0)\right].$$

After training, new global samples from $p_\phi(\mathbf{x}_0) \approx p(\mathbf{x}_0)$ are generated by running the reverse process in Equation (3) starting from $\mathbf{x}_T \sim \mathcal{N}(\mathbf{0}, \mathbf{I})$. Due to the stochastic nature of the reverse process, particularly, the additive noise during each step, starting from two close initial noise vectors at step $T$ does not necessarily lead to close-by images on the image manifold. The next section describes our proposed method for local sampling on the image manifold.

## 3 BOOMERANG METHOD

Our method, Boomerang, allows one to locally sample a point $\mathbf{x}'_0$ on an image manifold $\mathcal{X}$ close to a point $\mathbf{x}_0 \in \mathcal{X}$ using a pretrained diffusion model $f_\phi$. Since we are mainly interested in images, we suppose that $\mathbf{x}_0$ and $\mathbf{x}'_0$ are images on the image manifold $\mathcal{X}$. We indicate how close to $\mathbf{x}_0$ we want $\mathbf{x}'_0$ to be by setting the hyperparameter $t_{\text{Boomerang}}$. We perform the forward process of the diffusion model $t_{\text{Boomerang}}$ times, from $t = 0$ to $t = t_{\text{Boomerang}}$ in Equation (1), and use $f_\phi$ to perform the reverse process from $t = t_{\text{Boomerang}}$ back to $t = 0$. If $t_{\text{Boomerang}} = T$ we perform the full forward diffusion and hence lose all information about $\mathbf{x}_0$; this is equivalent to simply sampling from the diffusion model. We denote this partial forward and reverse process as $B(\mathbf{x}_0, t_{\text{Boomerang}}) = \mathbf{x}'_0$ and call it *Boomerang* because $\mathbf{x}_0$ and $\mathbf{x}'_0$ are close for small $t_{\text{Boomerang}}$, which can be seen in Figure 1.

When performing the forward process of Boomerang, it is not necessary to iteratively add noise $t_{\text{Boomerang}}$ times. Instead, we simply calculate the corresponding $\alpha_{t_{\text{Boomerang}}}$ and sample from Equation (2) once to avoid unnecessary computations. The reverse process must be done step by step however, which is where most of the computations take place, much like regular (non-local) sampling of diffusion models. Nonetheless, sampling with Boomerang has significantly lower computational costs than regular sampling; the time required for Boomerang is approximately $\frac{t_{\text{Boomerang}}}{T}$ times the time for regular sampling. Moreover, we can use Boomerang to perform local sampling along with faster sampling schedules, e.g., sampling schedules that reduce sampling time by 90% (Kong & Ping, 2021) before Boomerang is applied. Pseudocode for the Boomerang algorithm is shown in Algorithm 1. SDEdit (Meng et al., 2022) is an image editing method based on pretrained diffusion models, which uses a similar algorithm as above. This method is specific to image editing and is not used in the context of local sampling. To the best of our knowledge, Boomerang is the first work which uses generative models for local sampling on image manifolds.

---

**Algorithm 1** Boomerang local sampling, given a diffusion model $f_\phi(\mathbf{x}, t)$

---

**Input:** $\mathbf{x}_0, t_{\text{Boomerang}}, \{\alpha_t\}_{t=1}^{T}, \{\beta_t\}_{t=1}^{T}$
**Output:** $\mathbf{x}'_0$
   $\boldsymbol{\epsilon} \leftarrow N(\mathbf{0}, \mathbf{I})$
   $\mathbf{x}'_{t_{\text{Boomerang}}} \leftarrow \sqrt{\alpha_{t_{\text{Boomerang}}}} \mathbf{x}_0 + \sqrt{1 - \alpha_{t_{\text{Boomerang}}}} \boldsymbol{\epsilon}$
   **for** $t = t_{\text{Boomerang}}, ..., 1$ **do**
      **if** $t > 1$ **then**
         $\bar{\beta}_t = \frac{1 - \alpha_{t-1}}{1 - \alpha_t} \beta_t$
        $\boldsymbol{\eta} \sim N(\mathbf{0}, \bar{\beta}_t \mathbf{I})$
      **else**
        $\boldsymbol{\eta} = \mathbf{0}$
      **end if**
      $\mathbf{x}'_{t-1} \leftarrow f_\phi(\mathbf{x}'_t, t) + \boldsymbol{\eta}$
   **end for**
   **return** $\mathbf{x}'_0$

---

We present a quantitative analysis to measure the variability of Boomerang-generated images as $t_{\text{Boomerang}}$ is varied. As an expression of this variability, we consider the conditional distribution of samples generated through the Boomerang procedure conditioned on a noisy input image at step $t_{\text{Boomerang}}$, i.e., $p_\phi(\mathbf{x}'_0 \mid \mathbf{x}_{t_{\text{Boomerang}}})$. According to Bayes' rule, we relate this distribution to the distribution of noisy images at step $t_{\text{Boomerang}}$ of the forward process,

$$
\begin{aligned}
p_\phi(\mathbf{x}'_0 \mid \mathbf{x}_{t_{\text{Boomerang}}}) &\propto p_\phi(\mathbf{x}_{t_{\text{Boomerang}}} \mid \mathbf{x}'_0) p(\mathbf{x}'_0) \\
&\approx q(\mathbf{x}_{t_{\text{Boomerang}}} \mid \mathbf{x}'_0) p(\mathbf{x}'_0) \\
&= \mathcal{N}\left( \sqrt{\alpha_{t_{\text{Boomerang}}}} \mathbf{x}'_0, (1 - \alpha_{t_{\text{Boomerang}}}) \mathbf{I} \right) p(\mathbf{x}'_0).
\end{aligned}
\tag{7}
$$

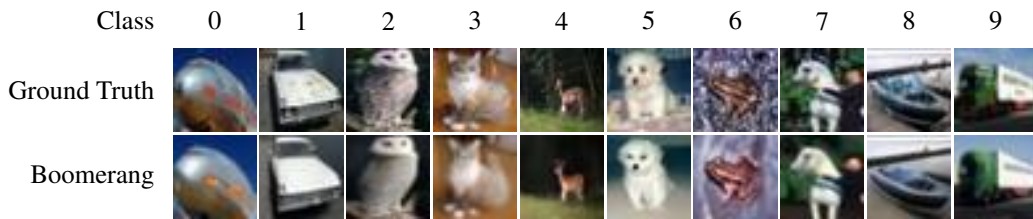

Figure 2: Using Boomerang on CIFAR-10 to change the visual features of images. These images were created with a FastDPM Model.

The second line in the expression above assumes that by training the diffusion model via the loss function in Equation (6), the model will be able to reverse the diffusion process at each step of the process. The last line in the equation above, which follows from Equation (2), suggests that the density of Boomerang-generated images is proportional to the density of a normal distribution with covariance $(1 - \alpha_{t_{\text{Boomerang}}})\mathbf{I}$ times the clean image density $p(\mathbf{x}_0')$, which is the same as the original clean image density. The resulting density will have very small values far away from the mean of the normal distribution. In addition, the high probability region of $p_\phi(\mathbf{x}_0' \mid \mathbf{x}_{t_{\text{Boomerang}}})$ grows as $1 - \alpha_{t_{\text{Boomerang}}}$ becomes larger. This quantity monotonically increases as $t_{\text{Boomerang}}$ goes from one to $T$ since $\alpha_t = \prod_{i=1}^{t}(1 - \beta_i)$ and $\beta_i \in (0, 1)$. As a result, we expect the variability in Boomerang-generated images to increase as we run Boomerang for larger $t_{\text{Boomerang}}$ steps.

Since Boomerang depends on a pretrained diffusion model $f_\phi$, it does not require the user to have access to significant amount of computational resources or data. This makes Boomerang very accessible to practitioners and even everyday users who don't have specialized datasets to train a diffusion model for their specific problem; they just need to find a diffusion model which is trained on diverse enough images, such as Stable Diffusion (Rombach et al., 2022). The main limitation of using Boomerang, however, is that the practitioner must find a diffusion model which models the image manifold well. If $f_\phi$ does not generate realistic images, then the output of Boomerang will also suffer, as we have seen empirically. However, this is becoming less and less of a problem with the advancement of diffusion models in image synthesis tasks. Overall, our Boomerang method allows local sampling on image manifolds without requiring significant amounts of computational resources or data.

## 4 APPLICATION 1: ANONYMIZATION OF DATA

### 4.1 DATA AND MODELS

In the following experiments we use the CIFAR-10 (Krizhevsky et al., 2009), FFHQ (Karras et al., 2019), and ILSVRC2012 (ImageNet) (Russakovsky et al., 2015) datasets. For the ImageNet experiments, we use a 200-class subset of ImageNet which we call ImageNet-200; these are the 200 classes that correspond to Tiny ImageNet (Russakovsky et al., 2015). We use Boomerang with the Stable Diffusion (Rombach et al., 2022), Patched Diffusion (Luhman & Luhman, 2022),[1] and FastDPM (Kong & Ping, 2021)[2] models, with some comparisons to the recent StyleGAN-XL (Sauer et al., 2022). For the FastDPM model we use Boomerang on the STEP DDPM sampler with $S = 100$ steps out of the original 1000 steps.

### 4.2 ANONYMIZATION

Recent work has proved that, not only are overparameterized models (such as deep networks) vulnerable to membership inference attacks (Shokri et al., 2017), but also that their vulnerability increases as their number of parameters increases (Tan et al., 2022). Such attacks attempt to recover potentially sensitive training data (e.g., medical data, financial information, and private images) given only access to a machine learning model (e.g., a classifier) trained on that data. Local sampling

---

[1] We use this repository.
[2] We use this repository.

| Class | 0 | 32 | 67 | 142 | 183 |

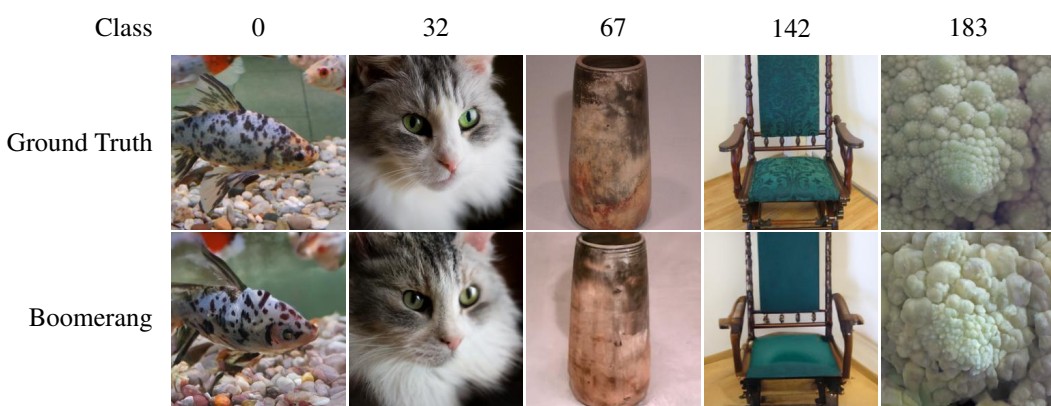

Figure 3: Using Boomerang on ImageNet-200 to change the visual features of images. These images were created with a Patched Diffusion Model.

presents a potential tool towards privacy-preserving machine learning by enabling the generation of data points that are similar to real (i.e., sensitive) data, while being dissimilar enough to existing data so as to not provide any true sensitive information, in the case of a membership inference attack. The Boomerang algorithm not only enables user-friendly local sampling for anonymization, but also enables the use of pre-trained and popular diffusion models in doing so, such as Stable Diffusion.

Stable Diffusion (Rombach et al., 2022) is a promising venue for Boomerang image anonymization, capable of both image feature modification and image style transfer, having been trained on a vast corpus of images containing a wide variety of images and styles (Schuhmann et al., 2021). Unlike other diffusion models, Stable Diffusion uses a Latent Diffusion Model (LDM) to conduct all forward and reverse diffusion processes solely in a latent space. As such, the Boomerang algorithm, when used on Stable Diffusion, first projects an image into the latent space using the LDM encoder, then adds noise to the latent vectors, next iteratively denoises the latent vectors via the LDM reverse process, and finally re-projects the denoised latent vectors back into the image space using the LDM decoder. Figure 1 (and Figures 5 to 7 in Appendix A.1) showcase the ability of Boomerang and Stable Diffusion to produce not only similar (i.e., anonymous) variants of an initial image, but also dissimilar variants that share certain features (e.g., a cat resembling a person, or a bathroom resembling a bedroom).

Additionally, Boomerang can be used to make entire datasets anonymized to varying degrees controlled by the hyperparameter $t_{\text{Boomerang}}$. Anonymization with people's faces, for example, makes intuitive sense, but anonymized images in a more general context, such as ImageNet, is less clearly defined. In this work, an image $\mathbf{x}_0$ is anonymized to $\mathbf{x}_0'$ if the features of each image are visibly different so that an observer would guess that the two images are of different objects. For each diffusion model, we pick $t_{\text{Boomerang}}$ so that the anonymized images are different, but not drastically different from the original dataset images, for CIFAR-10 and ImageNet-200. These images are shown in Figures 2 and 3. One potential alternative to using Boomerang-anonymized data is to use completely synthetic data. The state-of-the-art generative model for image generation of both ImageNet and CIFAR-10 is StyleGAN-XL. We compare our anonymized data to training a classifier with completely synthetic data.

We show in Table 1 that training a classifier on these anonymized images achieves a performance between using real images and using the state-of-the-art ImageNet generated images from StyleGAN-XL. This makes sense because, as we increase $t_{\text{Boomerang}}$, we obtain samples that are more similar to completely synthetic images. Therefore, anonymization of datasets is a better alternative for preserving the identity of data than using generated data.

## 5 APPLICATION 2: DATA AUGMENTATION

Data augmentation for image classification is essentially done by sampling points on the image manifold $\mathcal{X}$ near the training data. There are typical augmentation techniques, such as using random

Table 1: Using Boomerang-generated data for data augmentation increases test accuracy of CIFAR-10 and ImageNet-200 classification tasks. Using just the "anonymized" boomerang dataset for classification performs between using real data and using state-of-the-art completely synthetic StyleGAN-XL data.

| Classification Task | Training data | Top-1 Test Accuracy | Top-5 Test Accuracy |
|---|---|---|---|
| CIFAR-10 | CIFAR-10 data | 87.8% | |
| CIFAR-10 | StyleGAN-XL generated data | 81.5% | |
| CIFAR-10 | Boomerang (anonymized) data (ours) | 84.4% | |
| CIFAR-10 | CIFAR-10 + Boomerang DA (ours) | **88.4%** | |
| ImageNet-200 | ImageNet-200 data | 66.6% | 85.6% |
| ImageNet-200 | StyleGAN-XL generated data | 50.2% | 73.0% |
| ImageNet-200 | Boomerang (anonymized) data (ours) | 61.8% | 83.4% |
| ImageNet-200 | ImageNet-200 + Boomerang DA (ours) | **70.5%** | **88.3%** |

image flips and crops, which exploit symmetry and translation invariance properties of images. Although there are many techniques for data augmentation, they mostly involve modifying the training data in ways which make the new data resemble the original data while still being different, i.e., they attempt to perform local sampling on the image manifold.

Due to the intrinsic computational costs of diffusion models, we generate the augmented data before training instead of on-the-fly generation during training. We pick $t_{\text{Boomerang}}$ to be large enough to produce differences between the original dataset and the Boomerang-generated dataset, as shown in Figures 2 and 3. We then randomly choose to use the training data or the Boomerang-generated data with probability $0.5$ at each epoch. We use ResNet-18 (He et al., 2016) for our experiments.

We show that using Boomerang data augmentation on CIFAR-10 classification increases test accuracy from 87.8% to 88.4%. For CIFAR-10 we pick $t_{\text{Boomerang}} = 40$ and train for 100 epochs with a multistage learning rate scheduler which reduces the learning rate by a factor of 10 at epochs 30, 60, and 80.[3] We train using stochastic gradient descent with an initial learning rate of 0.1, Nesterov momentum of 0.9, and weight decay of $5\mathrm{e}{-4}$. Since CIFAR-10 already has a significant amount of samples per class (5,000) we see a small benefit from using Boomerang data augmentation.

We also perform data augmentation with ImageNet-200, described in Section 4.1, raising our test accuracy from 66.6% to 70.5%. We pick $t_{\text{Boomerang}} = 75$ and train for 90 epochs using the standard PyTorch (Paszke et al., 2019) training script[4] with a few modifications. We removed the standard data augmentation of random resized cropping and random flips and instead resize each image so that its smallest dimension is 256 followed by a centered crop to $224 \times 224$ in order to see if our Boomerang data augmentation would be beneficial. Since ImageNet-200 has much fewer samples than CIFAR-10 per class (approximately 1,300) and the classification task is harder, the greater increase in performance due to our data augmentation makes sense.

# 6  APPLICATION 3: SUPER-RESOLUTION

## 6.1  VANILLA BOOMERANG SUPER-RESOLUTION

To perform super-resolution with Boomerang, one first upsamples a low-dimensional image using any existing method (e.g., nearest-neighbor, linear interpolation), and then performs the Boomerang algorithm (See Algorithm 1) on this image to recover the "original" image in high dimensional space. The recovered image corresponds to a point on the image manifold that is "close" to the noisy, upsampled image; the distance between the two images is controlled by $t_{\text{Boomerang}}$.

While others such as Saharia et al. (2022b) and Rombach et al. (2022) have used diffusion models to perform well on super-resolution tasks, Boomerang has two key advantages to vanilla diffusion

---

[3]We use code from this repository to implement the CIFAR-10 experiments.
[4]For our ImageNet-200 training we used the standard PyTorch training script.

Initial 8x DS Image $\qquad$ $n_{\text{cascade}} = 5$

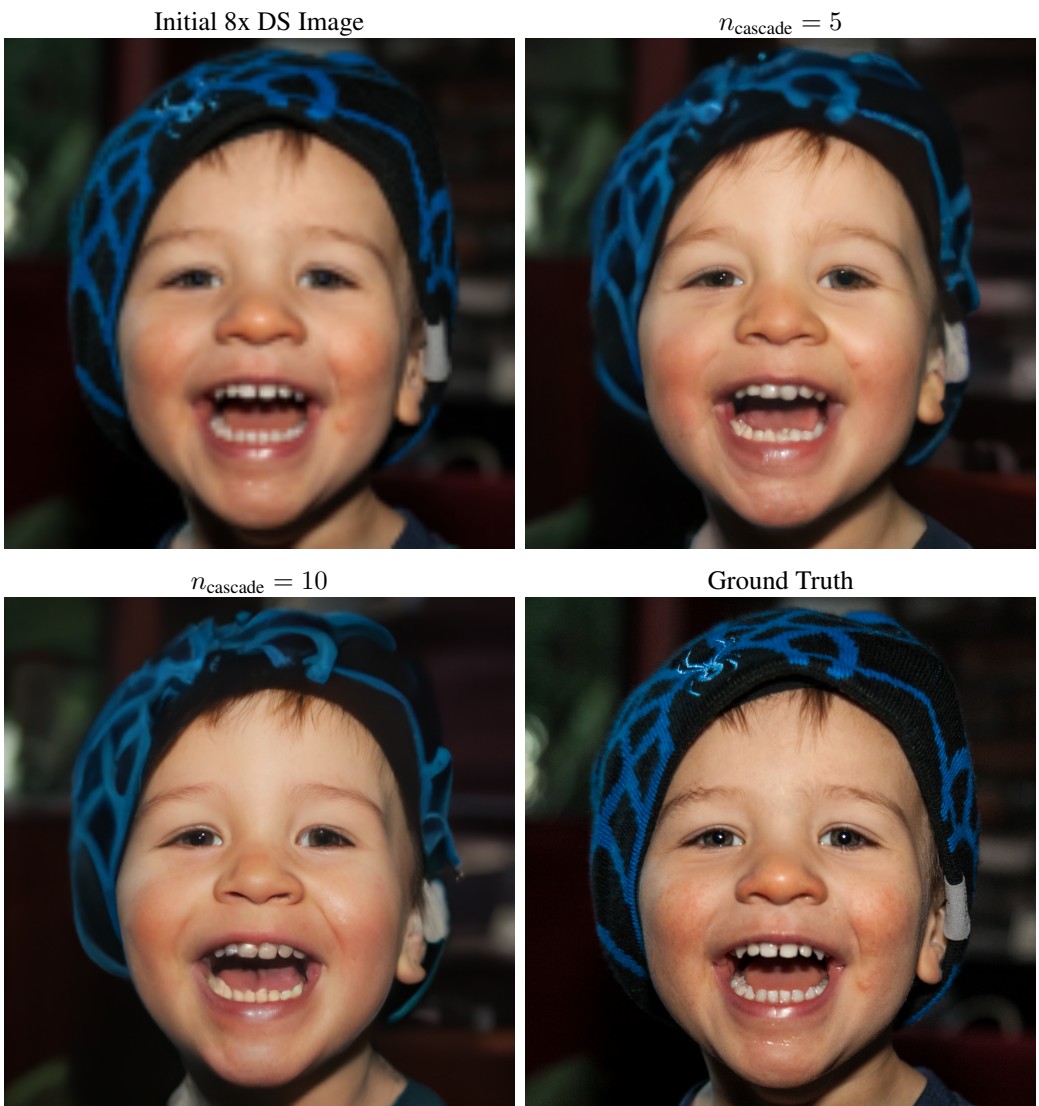

$n_{\text{cascade}} = 10$ $\qquad$ Ground Truth

Figure 4: Cascaded Boomerang super-resolution; $t_{Boomerang} = 50$

models. The first is that Boomerang's local sampling keeps the output "close" to the input on the image manifold. The second is that adjusting $t_{\text{Boomerang}}$ allows easy "tuning" of how aggressively the super-resolution enhancement is applied. This means that *the same pretrained network* can perform super-resolution on images of different scales: the more downsampled or degregated the image, the larger the value of $t_{\text{Boomerang}}$ one may choose to use to fill in more details of the final image. Furthermore, by varying $t_{\text{Boomerang}}$ for different passes of the same input image, one can choose which image has the best detail/variance tradeoff. This can be seen in Figure 8 in Appendix A.1, wherein more aggressive super-resolution improves clarity at the cost of distance from the original image. Empirical tests showed that setting $t_{\text{Boomerang}} \approx 100$ on the Patched Diffusion Model produced a good balance between sharpness and the features of the original image, in some cases maximizing PSNR.

## 6.2 Cascaded Boomerang Super-resolution

As $t_{\text{Boomerang}}$ is increased, the variance of the generated images dramatically increases due to diffusion models' stochastic nature (noise is added to data at each step $t$, see Algorithm 1). As a result,

increasing $t_{\text{Boomerang}}$ enough causes the generated images to vary so much that they no longer resemble the input image at all (and thus we are not sampling from the image manifold as closely as we desire). Even for modest values of $t_{\text{Boomerang}}$, the large variance of added noise causes repeated upsampling attempts to differ significantly. In this section, we propose a simple method that allows the results of different upsampling attempts to be stabilized.

The cascade method describes repeated passes of an upsampled image through a diffusion network with a smaller value of $t_{\text{Boomerang}}$. If we denote the Boomerang upsampling method in the previous section on an input image $\mathbf{x}_{\text{ds}}$ to generate $\mathbf{x}_{\text{sr}} = B_{f_\phi}(\mathbf{x}_{\text{ds}})$, the method described here would be $\mathbf{x}_{\text{cascade}} = B_{f_\phi}(B_{f_\phi}(\dots(B_{f_\phi}(\mathbf{x}_{\text{ds}}))))$. We designate $n_{\text{cascade}}$ as the number of times we repeat Boomerang on the intermediate result. In addition to stabilizing independent super-resolution attempts, the cascade method allows users to iteratively choose the desired upsampling detail—simply stop repeating the cascade process once the desired resolution is achieved. An example of cascaded super-resolution with Boomerang is shown in Figure 4, with more details found in Figure 9.

Super-resolution is an essential inverse image problem (Freeman et al., 2002; Wang et al., 2021), and Boomerang is a quick and efficient method of performing super-resolution with diffusion networks. As we have shown, Boomerang allows the user to easily adjust the strength of the super-resolution by varying $t_{\text{Boomerang}}$. The same procedure and pretrained network can thus be used to perform super-resolution at any scale, for any image size smaller than the output of the diffusion network (in our emperical tests, the image dimensions were $1024 \times 1024$): for larger scaling factors, simply increase $t_{\text{Boomerang}}$ or cascade the result until one achieves the desired fidelity. This also avoids the issue of needing to train different networks for upsampling at different scales. As with the previous applications, Boomerang's generated images look realistic yet it requires no additional fine-tuning or training.

Interestingly, performing super-resolution using Boomerang with some diffusion models worked and with some it did not work. With Stable Diffusion, for example, empirical results had shown that the vanilla Stable Diffusion network would not introduce detail into blurred images except for large $t_{\text{Boomerang}}$, at which point the generated images no longer strongly resembled the ground truth images. This is likely due to the fact that noise is added in the latent space with Stable Diffusion but in the image space with Patched Diffusion. Therefore, the noise model will impact which kinds of inverse problems Boomerang will be effective on.

## 7 CONCLUSION

We have presented the Boomerang algorithm, which enables simple and efficient local image sampling via diffusion models on a single GPU, without any re-training or modifications to the model. We showed the applicability of Boomerang on various tasks, such as anonymization, data augmentation, and super-resolution. Future works include continued experiments of its efficacy for data augmentation, as well as applying the Boomerang algorithm to different domains of data, such as audio and text. Additionally, recent work has shown that diffusion models can work with non-stochastic transforms instead of additive Gaussian noise (Bansal et al., 2022; Daras et al., 2022), and evaluating the Boomerang algorithms with such diffusion models would provide further insight into the nature of local sampling using diffusion models.

## ACKNOWLEDGEMENT

This work was supported by NSF grants CCF-1911094, IIS-1838177, and IIS-1730574; ONR grants N00014-18-12571, N00014-20-1-2534, and MURI N00014-20-1-2787; AFOSR grant FA9550-22-1-0060; and a Vannevar Bush Faculty Fellowship, ONR grant N00014-18-1-2047.

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

# A APPENDIX

## A.1 BOOMERANG-GENERATED IMAGES VIA THE STABLE DIFFUSION MODEL

Here we present additional images created via the Boomerang method that indicate the evolution of the predicted image as we increase $t_{\text{Boomerang}}$. These images are generated via the pretrained Stable Diffusion model (Rombach et al., 2022) where instead of adding noise to the image space during the forward process it is added in the latent space. Figures 5–7 showcase this where the images on the bottom row show noisy latent variables whereas the ones on the top row indicate the Boomerang predictions with increasing amounts of added noise from left to right, except for the rightmost image, which is created by using an alternate prompt.

## A.2 VANILLA BOOMERANG SUPER-RESOLUTION

Here we present the result of image super-resolution using the vanilla Boomerang approach. Figure 8 illustrates the results for image super-resolution. The top-left image in this figure shows the low-resolution image, and the top-right and bottom-left images are the result of vanilla Boomerang 8x super-resolution when using $t_{\text{Boomerang}} = 100$ and $t_{\text{Boomerang}} = 150$, respectively. When compared with the high-resolution image in the bottom-right corner of Figure 8 we observe that the resulting image with $t_{\text{Boomerang}} = 100$ is plausible while the result with $t_{\text{Boomerang}} = 150$ seems high-resolution, but is inconsistent compared to the original image.

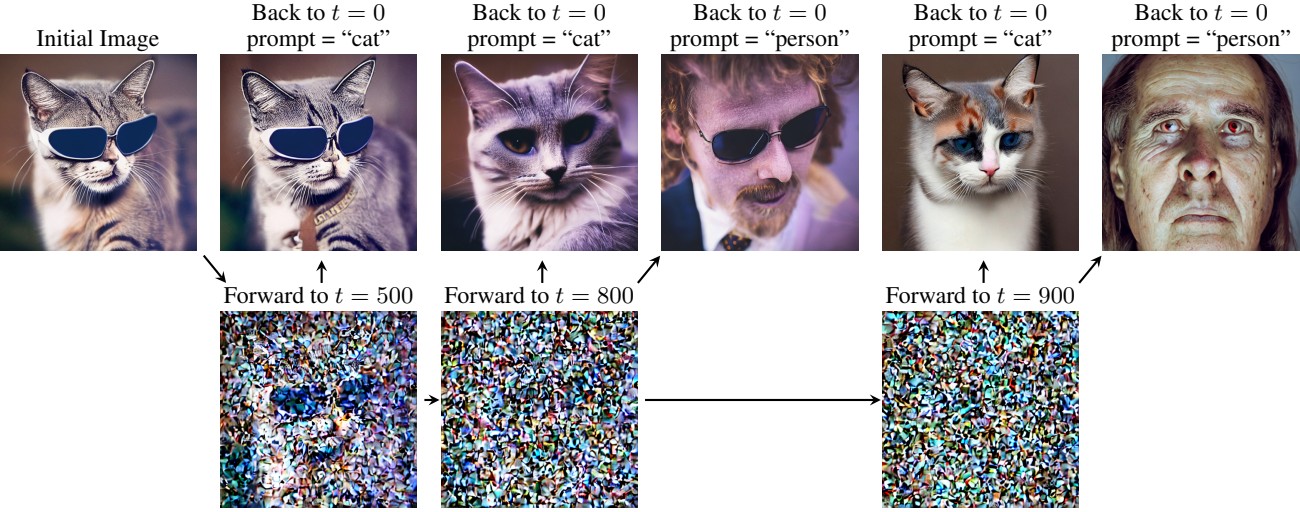

Figure 5: The Boomerang method using Stable Diffusion ($T = 1000$), as in Figure 1, with an image of a cat. Note how, as $\frac{t_{\text{Boomerang}}}{T}$ approaches 1, the content of Boomerang-generated images *strays* further away from the starting image.

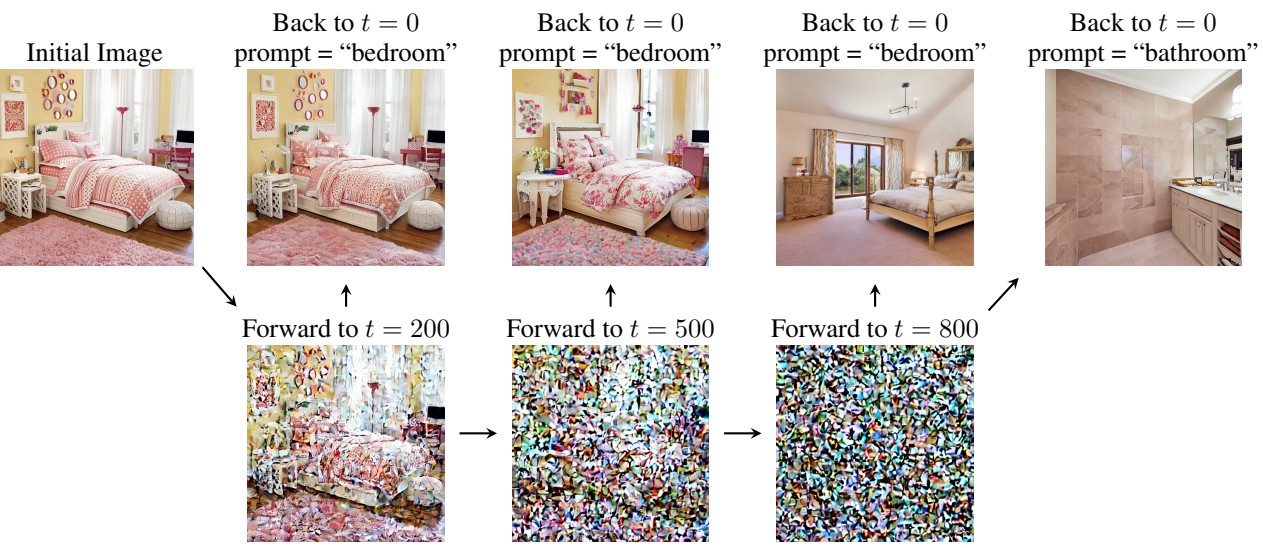

Figure 6: The Boomerang method using Stable Diffusion, as in Figure 1, with an image of a bedroom.

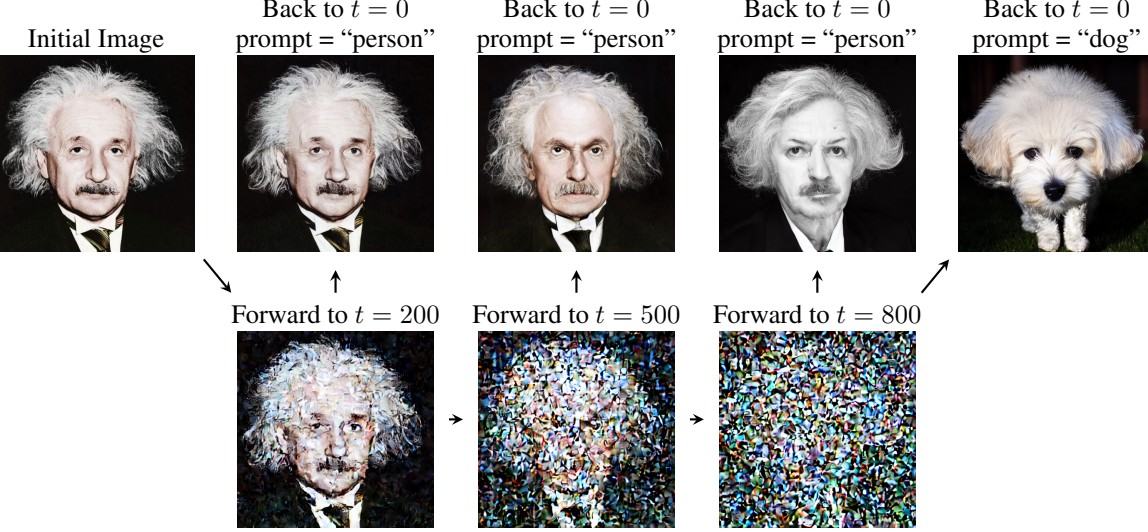

Figure 7: The Boomerang method using Stable Diffusion, as in Figure 1, with an image of Albert Einstein.

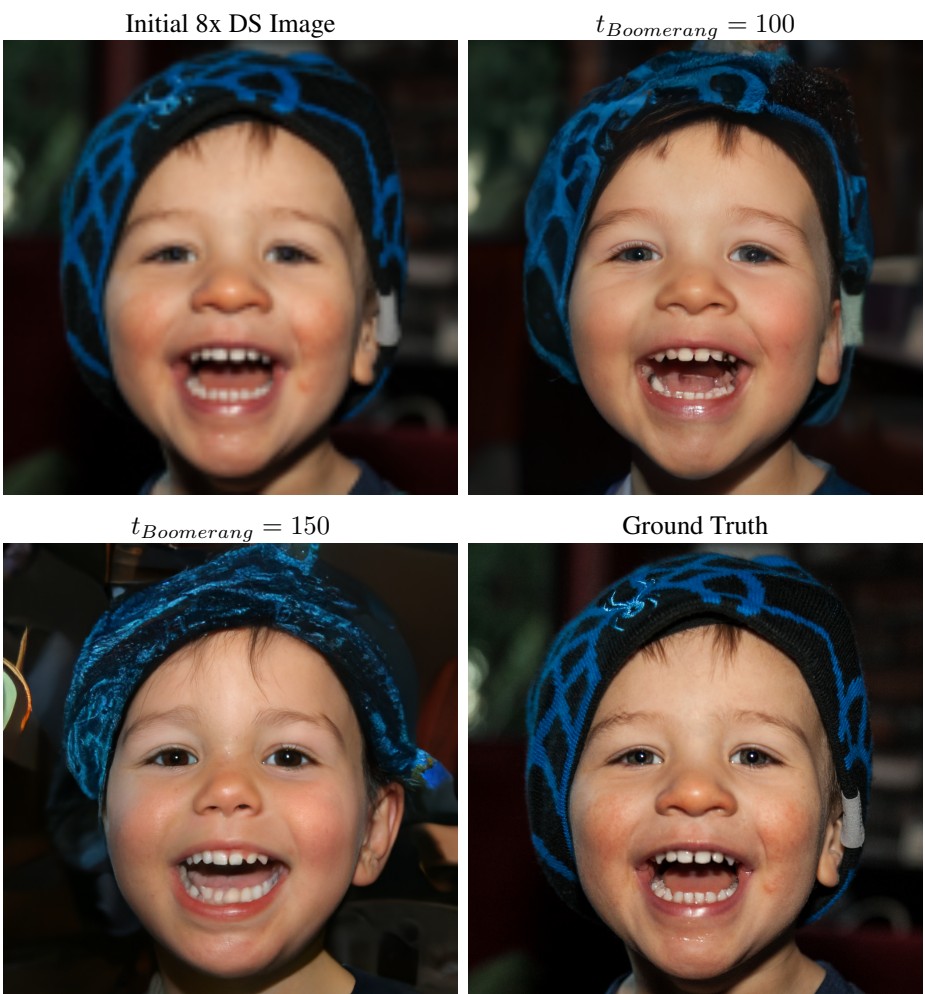

Figure 8: Vanilla super-resolution with Boomerang.

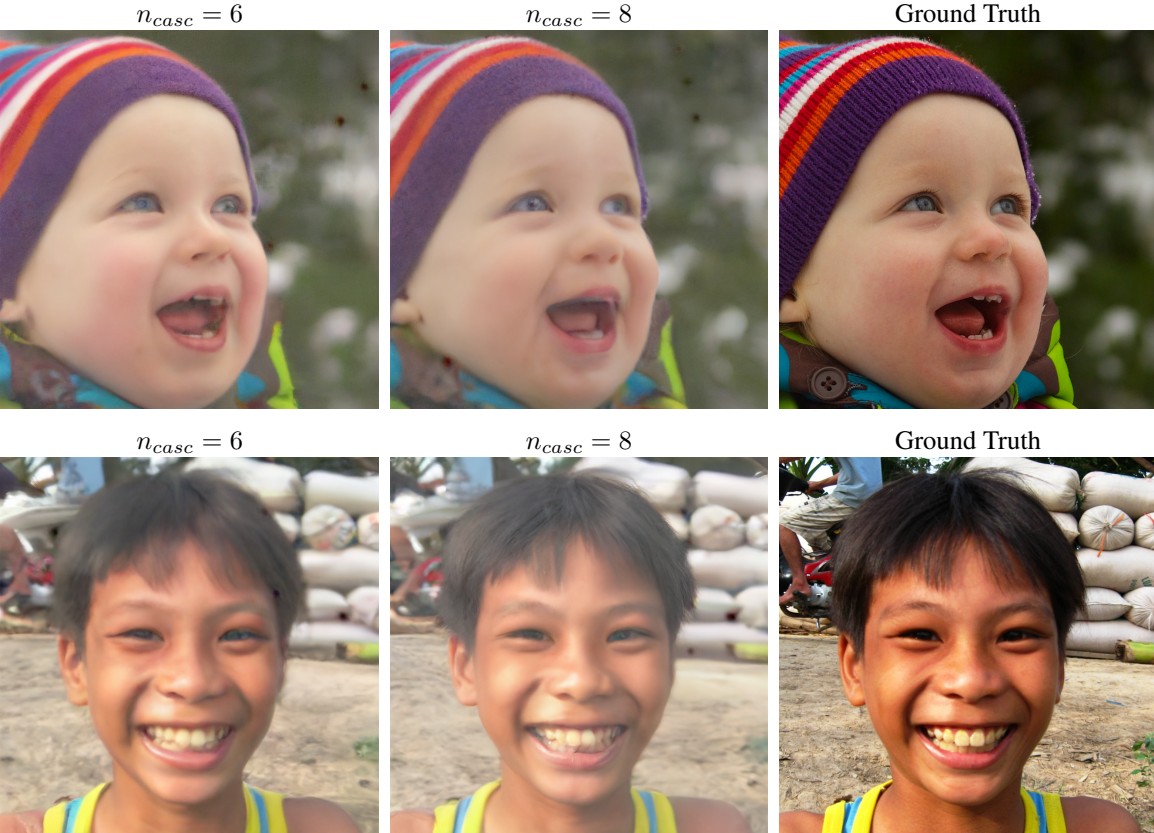

Figure 9: Cascaded boomerang examples with superresolution. For the top row, notice how the best quality image is seen after 6 cascade steps, with $t_{Boomerang} = 50$. After 8 cascades, details such as the teeth begin to be removed. On the bottom row, however, $n_{casc} = 8$ provides better results and more detail compared to previous steps. This shows the value in introducing the cascade method: different images may have better results with more cascade steps than others

