# OpenReview forum: "Boomerang: Local sampling on image manifolds using diffusion models"
_ICLR.cc/2023/Conference — Submitted to ICLR 2023_

### Official Review · Reviewer_z9Wv · 2022-10-24

**Confidence:** 4
**Correctness:** 3
**Technical Novelty And Significance:** 1
**Empirical Novelty And Significance:** 1
**Recommendation:** 3

**Clarity, Quality, Novelty And Reproducibility:**

- Clarity:The paper is well written and easy to understand.
- Novelty: the proposed approach for inversion was already introduced in the original DDPM paper.
- Reproducibilty: As far as this reviewer can tell all the details for reproduction are given in the paper and results are shown on standard benchmark datasets.

**Strength And Weaknesses:**

Strengths:
- Successfully inverting a sample in diffusion models is an interesting problem that deserves more attention from the community.
- The proposed approach is technically correct.

Weakness:
- The technical contributions are not novel, the same approach was introduced in the original DDPM paper https://arxiv.org/abs/2006.11239  (see appendix for interpolation results).
- The empirical evaluation is rather limited. It is hard to evaluate the value of the contribution with qualitative results on data augmentation.

**Summary Of The Paper:**

This submission proposes a method for inverting samples in diffusion generative models. The proposed approached simply applies the forward diffusion process for a given number of steps and then samples from the resulting latent. The approach is evaluated on 3 different problems: privacy-preserving reconstruction, data augmentation and super-resolution.

**Summary Of The Review:**

This submission tackles an interesting problem (eg. Inversion in diffusion models). The paper is clearly written and easy to understand. The main drawback of the proposed approach is the novelty of it, since an equivalent method was already introduced in the original DDPM paper.  Furthermore, the qualitative evaluation is not complete and makes it hard to validate the overall value of the contribution

---

> ### Author Response · Authors · 2022-11-19
> **Official Response to Reviewer z9Wv**
>
> Thanks for your review! We appreciate the in-depth discussion of DDPM.
>
> We agree with you that the DDPM paper does some interesting things which are related to our Boomerang algorithm. One of these similarities is the investigation of the stochasticity of the reverse process by freezing the reverse process at step $t$ and resuming it multiple times, starting from step $t$. According to the authors, as the freezing step $t$ (equivalent to $t_{\text{boomerang}}$) decreases, the stochasticity of the reverse process decreases as well. We rely on this property of the diffusion reverse process in order to develop a framework for local sampling. Our approach differs from the process described by the DDPM paper in that Boomerang provides images which are near (on the manifold) to an existing image, unlike the DDPM approach, which begins with a random realization of a Gaussian distribution, resulting in local samples around an arbitrarily, uncontrolled region of the manifold.
>
> The second similarity with the DDPM paper is regarding their interpolation examples, in which the reverse process at stage $t$ is used to decode linearly interpolated latent codes. While some of the decoded images could be considered as local samples, this approach is not inherently a local sampling technique and requires a second image with which to perform interpolation, hence not being the same approach as Boomerang, as the reviewer claims. Therefore, the DDPM interpolation method would not be able to reap any of the benefits of our Boomerang local sampling approach that we discussed in our note on novelty. We hope this clears up any confusions about the similarities of Boomerang and the work done in the DDPM paper.
>
> We agree that having only qualitative results would be rather unconvincing, which is why we provided quantitative results on data augmentation in the original submission of the paper. We ask the reviewer to look at Table 1 showing that using Boomerang-generated data for data augmentation, without any other image augmentation techniques, increases test accuracy of CIFAR10 and ImageNet-200 classification tasks. Hopefully this alleviates your concerns about the qualitative nature of data augmentation.
>
> We have provided a Colab demo showcasing usage of our algorithm with the Stable Diffusion network in the caption of Figure 1.

---

### Official Review · Reviewer_Wv4p · 2022-10-25

**Confidence:** 4
**Clarity, Quality, Novelty And Reproducibility:** The paper is clearly written. The wor…
**Correctness:** 3
**Technical Novelty And Significance:** 1
**Empirical Novelty And Significance:** 1
**Recommendation:** 3

**Strength And Weaknesses:**

I think what this paper proposes is known and not new. Particularly, it seems that the Boomerang Algorithm is exactly the same as the SDEdit paper: https://arxiv.org/abs/2108.01073. The latter is not cited and the differences are not discussed, but I really don't understand if there is anything new here.

The super-resolution application seems interesting. However, it would be good to see MSE and FID scores and compare it against other approaches - e.g. compare with super-resolution with GANs.

For the data augmentation application, I think that the comparison with StyleGAN-XL is unfair. Unconditional generation with StyleGAN-XL is expected to not match the quality of the original data samples. But data augmentation with Boomerang is not unconditional. Dataset samples are only slightly augmented. A more fair comparison would be to do inversion with StyleGAN-XL and add the inverted images to the dataset. There are many methods for controlling the fidelity of the inverted image to the given input which would be the equivalent of the t_boomerang knob.


**Summary Of The Paper:**

The paper proposes a method for generating variations of an existing image with diffusion models. The key idea is to add noise to the given image and then reconstruct it using denoising diffusion models. As you add more noise, the variability increases.

**Summary Of The Review:**

The paper introduces a method for producing variations of a given image. I believe that this method is already known. Hence, I recommend rejection.

---

> ### Author Response · Authors · 2022-11-19
> **Official Response to Reviewer Wv4p**
>
> Thank you for bringing to our attention the SDEdit paper, which has similarities with our own work which we have added to our paper. The algorithm part of the SDEdit paper is similar to our own but there are several distinct differences:
>
> 1) SDEdit is different from Boomerang: The SDEdit algorithm performs the full forward and reverse process with modified noise schedule, while we only do a partial forward and reverse process without modifying the noise schedule. This makes Boomerang algorithmically different and much more computationally efficient than SDEdit. For example, if our $t_{\text{boomerang}} = 50$ out of $250$, we would actually be only performing $\frac{50}{250} = 20$% of the computations which saves on time and computational resources.
>
> 2) Image editing is not local sampling: As mentioned in the note on novelty, the SDEdit algorithm’s goal is to edit photorealistic images or create them from sketches and other non-natural images. In contrast, Boomerang performs local sampling on image manifolds, meaning that we start with a natural image and produce nearby natural images on the learned manifold. SDEdit does not do anonymization of data, data augmentation, or super-resolution and hence our work is completely out of scope of the SDEdit work.
>
> Thank you for your interest in our super-resolution work! We really like the idea of using FID to compare super resolution samples as we are not performing super-resolution on a fixed grid, which would be better suited for MSE. However, see the note on super-resolution comparisons explaining why we don’t compare to super-resolution models using GANs.
>
> With regards to data augmentation, we understand your concerns about the comparison with StyleGAN-XL if it was unconditional. However, the pretrained StyleGAN-XL model provides conditional samples from Cifar10 and ImageNet datasets (as opposed to unconditional sampling for the FFHQ and Pokémon datasets). For more details, see the official Github repository: https://github.com/autonomousvision/stylegan_xl.  We hope that this alleviates your concern about the fairness of the data augmentation comparisons.

---

### Official Review · Reviewer_ydwA · 2022-10-26

**Confidence:** 3
**Correctness:** 4
**Technical Novelty And Significance:** 3
**Empirical Novelty And Significance:** 3
**Recommendation:** 8

**Clarity, Quality, Novelty And Reproducibility:**

*Quality:* The work appears to be technically sound, and the claims of the paper are well supported. Moreover, the authors clearly communicate the limitations of the approach.

*Clarity:* The paper is well organized and clearly written.

*Originality:* The work appears to be an original use of an existing tool.



**Strength And Weaknesses:**

*Strengths:* This paper introduces a method for locally sampling the data manifold in a diffusion model. The proposed method is intuitive and the authors have demonstrated the applicability of the approach to several tasks. Moreover, the method proposed can be used with pretrained models, on a single GPU making it accessible to the larger community.

*Possible typos:*
- “better alternative to preserving” → better alternative for preserving



**Summary Of The Paper:**

The authors introduce a method for local sampling in diffusion models. This is achieved by leveraging the stochasticity of the diffusion models. An input image is propagated through the forward process for t steps then through the reverse process for t steps. In the forward process each step produces an output that is the linear combination of the input and a noise vector – the combination coefficient for the input is determined by the choice of variance for the noise. The reverse process is governed by a learned transition function. The authors demonstrate the utility of their local sampling procedure for anonymization of data, dataset augmentation, and image super-resolution.

**Summary Of The Review:**

I think the authors have identified a simple and powerful approach for local sampling in diffusion models. The paper is clearly written, and appears detailed enough that others could reproduce the findings.

---

> ### Author Response · Authors · 2022-11-19
> **Official Response to Reviewer ydwA**
>
> Thank you so much for your review! We agree that Boomerang leverages an existing principle towards the new, generally applicable task of local sampling on manifolds; such a task has broad implications for anonymization, data augmentation, and super-resolution. To further improve the accessibility of Boomerang to the broader scientific community, we have provided a Colab demo showcasing usage of our algorithm with the Stable Diffusion network in the caption of Figure 1.

---

### Official Review · Reviewer_qWcF · 2022-11-03

**Confidence:** 4
**Correctness:** 2
**Technical Novelty And Significance:** 1
**Empirical Novelty And Significance:** 2
**Recommendation:** 3

**Clarity, Quality, Novelty And Reproducibility:**

Clarity and Quality: The paper is more-or-less well-written.

Novelty: The work under consideration could be characterized as a description of a nice scenario of diffusion models application.

Reproducibility: The code not provided in the supplementary materials


**Strength And Weaknesses:**

Strength: The idea is quite simple and natural and could be applied for arbitrary pretrained diffusion models.

Weaknesses:
In general. My main concern about the paper is that the proposed approach is too incremental and kind of obvious. It is really nice that such a simple method helps to solve rather important problems listed in the application sections but from my point of view it is not enough reason for the paper under consideration to be published in top A* conference.

In particular.
1) No comparison with competitive super-resolution methods is provided. Therefore it is difficult to estimate the usefulness of the Boomerang approach in this application.
2) No quantitative comparison of the vanilla Boomerang and cascaded Boomerang super-resolution is provided. It is not clear, if the “casacadeness” actually improves samples quality
3) Table 1 doesn’t seem to be convincing and requires clarifications. The reported accuracies of ResNet-18 model trained on standard Cifar-10 (and, probably ImageNet-200) data are actually not sota accuracies for ResNet-18 and corresponding datasets (see, https://github.com/kuangliu/pytorch-cifar for example). Therefore they are just particular accuracies obtained in particular training loops. This observation gives rise to the questions like as follows: Can we just train ResNet-18 for 200 (180 for ImageNet) epochs and achieve the competitive accuracy compared to Boomerang-augmented dataset cases? Can we achieve competitive performance using standard augmentation techniques?


**Summary Of The Paper:**

The paper under consideration proposes a “Boomerang” approach for image variability-related applications. The idea is quite simple. Given a standard diffusion model:
- run forward diffusion for several times resulting in noised image
- run backward diffusion for several times resulting in an image which is controllably different from the original one.

The authors demonstrate the application of the aforementioned method in several applications.


**Summary Of The Review:**

Summary Of The Review: The proposed approach is quite straightforward and incremental. The applications are nice and practically interesting but require further investigation.

---

> ### Author Response · Authors · 2022-11-19
> **Official Response to Reviewer qWcF**
>
> We thank you for your detailed review.
>
> Although our method for local sampling is simple, no previous work has applied diffusion model inversion for broadly important challenges in signal processing (such as data augmentation, anonymization, and super-resolution) without specialized training or fine-tuning procedures. See the note on novelty for a more in-depth explanation of Boomerang’s novelty and utility.
>
> We agree that a more rigorous comparison between vanilla and cascaded Boomerang for super-resolution is needed, so we have added a qualitative comparison to the paper in Figure 9.  We found that while PSNRs tended to be higher for cascaded results, they were not indicative of the overall image quality, so we refrained from using this metric in the paper.  Instead, we demonstrate the benefits of cascaded Boomerang with several example images shown in Figure 9 in the supplementary materials.
>
> The reviewer is also concerned that we are using a suboptimal training procedure, which is an important point. Our results for CIFAR-10 do not match SOTA results for ResNet-18 as is shown by the link that the reviewer provided. This is true, however those SOTA results and our results cannot be fairly compared because they use data augmentation techniques which we do not. In our baseline methods (first and fifth rows of Table 1 in the paper) we do not use ANY data augmentation so that we can study how the Boomerang data augmentation affects the test accuracy.
>
> We trained our CIFAR-10 model using the cutout repository (https://github.com/uoguelph-mlrg/Cutout) which achieves 96.01% accuracy with ResNet-18 on CIFAR-10 with several different data augmentation techniques. For our ImageNet-200 training we used the standard PyTorch training script (https://github.com/pytorch/examples/blob/main/imagenet/main.py).
>
> So, to answer the reviewer’s questions:
>
> Can we just train ResNet-18 for 200 (180 for ImageNet) epochs and achieve the competitive accuracy compared to Boomerang-augmented dataset cases? Increasing the training epochs did not change the loss whatsoever as these two scripts are already very well optimized by the authors of the Cutout paper and by the PyTorch team.
>
> Can we achieve competitive performance using standard augmentation techniques?
> Yes definitely! If you add standard augmentation techniques then you would have essentially SOTA performance. However, our claim here is that adding Boomerang data augmentation is beneficial, not that it gives you SOTA results, as that is outside the scope of this paper.
>
> As for reproducibility, code using Stable Diffusion via the Boomerang algorithm is provided in the caption of Figure 1.

---

### Author Response · Authors · 2022-11-18
**A note on novelty**

We thank you all for the reviews of our paper, which we think is very interesting and novel. Since several of the reviews we received expressed concern about the novelty of our work, we would like to address them here. Specifically, our work is novel for the following three reasons:
1. To the best of our knowledge, our work is the first which proposes using generative models for local sampling on image manifolds. No other generative model that we have found is designed to take an arbitrary natural image and produce another similar image, starting from the image manifold and sampling nearby points. There are other algorithms, such as SDEdit, which project onto the image manifold from non-image points, but that is not the task that this work focuses on.
2. Our method is very general, and we can apply it to tasks such as anonymization of data, data augmentation, and superresolution. We have not found any other work which offers an algorithm for all three of these areas.
3. Our method is simple, computationally efficient, and requires no additional training, making it usable by anyone who has a single GPU.

---

### Author Response · Authors · 2022-11-19
**A note on super-resolution comparisons**

While we did not include a comparison between Boomerang super-resolution and other competitive super-resolution methods, a thorough search through modern super-resolution literature revealed that, to our knowledge, there does not exist any prior work with which Boomerang super-resolution could be fairly compared. In part, this is because the existing approaches are explicitly designed for the super-resolution problem, or involve embedding a pre-trained generative model into a super-resolution framework through inference-time training or optimization. The Boomerang super-resolution method, however, does not involve any further optimization and it only requires performing Boomerang sampling through a diffusion model pretrained on a relevant dataset. Additionally, Boomerang sampling circumvents the fragility of existing super-resolution models in the extreme downsampling regime [9] since diffusion models always return a point on the learned image manifold. Here we list recent competitive works in the field of super-resolution and briefly explain why each work would not produce a fair comparison with the Boomerang algorithm.  In particular, all GAN methods for super-resolution are specifically trained or conditioned on the downsampled image when the discriminator is trained, so none of the GAN-based methods we came across could be fairly compared to Boomerang.

**GLEAN: Generative Latent Bank for Large-Factor Image Super-Resolution**: This work [1] requires a specialized architecture trained around another pre-trained generative network, and is specifically designed for super-resolution.

**Deep Image Prior**:  This method [2] is untrained and does not learn an image manifold, unlike diffusion models, which do learn an image manifold. Therefore, comparisons with Boomerang would be unfair against the Deep Image Prior.

**SRGAN**: This network [3] is specifically designed for super-resolution: it uses a perceptual similarity metric, has low-dimensional images as latent variables, and is trained on a super-resolution task.

**ESRGAN: Enhanced Super-Resolution Generative Adversarial Networks**: This work [4] is trained specifically for super-resolution, and is thus as incomparable with the Boomerang algorithm as is its predecessor, SRGAN.

**Image Super-Resolution Using Deep Convolutional Networks**: This work [5], while foundational in the field of deep super-resolution, was trained specifically for super-resolution, unlike Boomerang sampling of diffusion models.

**Cross-Scale Internal Graph Neural Network for Image Super-Resolution**: This work [6] has a specialized architecture and is specifically trained for super-resolution.

**Cold Diffusion: Inverting Arbitrary Image Transforms Without Noise**: This work [7] has diffusion models which, unlike the almost all other publicly available diffusion models, were specifically trained using a downsampling forward process, which is more specialized to the task of super-resolution than Boomerang sampling of diffusion models.

**Image Super-Resolution via Iterative Refinement**: This work [8], unlike Boomerang sampling, was specifically trained to do image-conditioned superresolution, and has to go through the full reverse process (unlike Boomerang, which only has to compute a fraction of the reverse process).

---

> ### Author Response · Authors · 2022-11-19
> **References**
>
> [1] Chan, K. C. K. Wang, X., Xu, X., Gu, J., and Loy, C. C. (2021). GLEAN: generative latent bank for large-factor image super-resolution. Proceedings of the IEEE conference on computer vision and pattern recognition.
>
> [2] Ulyanov, D., Vedaldi, A., and Lempitsky, V. (2017). Deep image prior. arXiv preprint arXiv:1711.10925.
>
> [3] Christian L., Lucas T., Ferenc H., Jose C., Andrew C., Alejandro A., Andrew A., Alykhan T., Johannes T., Zehan W., and Wenzhe S. (2017). Photo-realistic single image super-resolution using a generative adversarial network. Proceedings of the IEEE conference on computer vision and pattern recognition.
>
> [4] Wang, X., Yu, K., Wu, S., Gu, J., Liu, Y., Dong, C., Loy, C. C., Qiao, Y., and Tang, X. (2018). ESRGAN: enhanced super-resolution generative adversarial networks. Proceedings of the european conference on computer vision workshops
>
> [5] Dong, C., Loy, C. C., He, K., and Tang, X. (2015). Image super-resolution using deep convolutional networks. arXiv preprint arXiv:1501.00092.
>
> [6] Zhou, S., Zhang, J., Zuo, W., and Loy, C. C. (2020). Cross-scale internal graph neural network for image super-resolution. Advances in neural information processing systems.
>
> [7] Bansal, A., Borgnia, E., Chu, H., Li, J. S., Kazemi, H., Huang, F., Goldblum, M., Geiping, J., and Goldstein, T. (2022). Cold diffusion: inverting arbitrary image transforms without noise. arXiv preprint arXiv:2208.09392.
>
> [8] Saharia, C., Ho, J., Chan, W., Salimans, T., Fleet, D. J., and Norouzi, M. (2022). Image super-resolution via iterative refinement. IEEE transactions on pattern analysis and machine intelligence.
>
> [9] Yu, X., and Porikli, F. (2016). Ultra-resolving face images by discriminative generative networks. Proceedings of the european conference on computer vision.

---

### Decision · Program_Chairs · 2023-01-20

**Decision:**

Reject

**Justification For Why Not Higher Score:**

If the paper can show how the proposed approaches can do something that are previously implausible, or can significantly improve previous methods, then the concerns on lack of technical novelty could be mitigated. However, the proposed approaches are not there yet.

**Justification For Why Not Lower Score:**

N/A

**Metareview: Summary, Strengths And Weaknesses:**

The paper provides several relatively straightforward application of a pretrained denoising diffusion model, and the performed experiments are not considered by the reviewers to be that exciting or challenging. The lack of technical novelty becomes the main reason to place the paper below the acceptance bar.